# Comprehensive Evaluation and Selection of Cardamom (*Elettaria cardamomum* (L.) Maton) Germplasm Using Morphological Traits

**DOI:** 10.3390/plants13192786

**Published:** 2024-10-04

**Authors:** Martha Patricia Herrera-González, Alejandra Zamora-Jerez, Rolando Cifuentes-Velasquez, Luis Andrés Arévalo-Rodríguez, Santiago Pereira-Lorenzo

**Affiliations:** 1Center for Agricultural and Food Studies, Universidad del Valle de Guatemala, Guatemala City 01015, Guatemalalaarevalo@uvg.edu.gt (L.A.A.-R.); 2Programa de Doctorado en Agricultura y Medio Ambiente para el Desarrollo, Universidade de Santiago de Compostela, 27001 Lugo, Spain; 3Center for Biotechnology Studies, Universidad del Valle de Guatemala, Guatemala City 01015, Guatemala; oazamora@uvg.edu.gt; 4Departamento de Producción Vegetal y Proyectos de Ingeniería, Escola Politécnica Superior de Enxeñaría Campus Terra, Universidade de Santiago de Compostela, Lugo 27001, Spain; santiago.pereira.lorenzo@usc.es

**Keywords:** cardamom germplasm, morphological traits, comprehensive evaluation, morphological composite value F

## Abstract

Cardamom (*Elettaria cardamomum* (L.) Maton) plays a crucial role in Guatemala’s agriculture, supporting local families and covering 169,429.29 ha (making it the world’s leading producer). Since its introduction to Guatemala in 1910, limited research has focused on unraveling the diversity and defining morphological traits critical for selecting excellent accessions. In this study, we examined 17 morphological traits across 288 accessions to identify key features associated with the germplasm. The comprehensive analysis employed principal component analysis, a morphological composite value (F-value), linear regression, and hierarchical clustering. The Shannon–Wiener diversity index ranged from 0.10 to 2.02, indicating the variation in diversity among traits. Principal component analysis and hierarchical clustering revealed six distinct germplasm groups. The comprehensive analysis facilitated the selection of 14 excellent accessions, and the regression equation incorporating criteria such as plant height, capsule color, panicle number per plant, panicle length, rhizome color, cluster number per panicle, cluster node length, and capsule number per cluster to identify cardamom germplasm. To develop a conservation strategy for the two putative foreign varieties (‘Malabar’ and ‘Mysore’/’Vazhukka’) introduced in Guatemala based on plant height, another 12 accessions were selected with a second comprehensive evaluation. This information offers insights into cardamom diversity for informed selection enhancing national utilization, productivity, and conservation.

## 1. Introduction

Cardamom (*Elettaria cardamomum* (L.) Maton), commonly known as small cardamom is a perennial herbaceous plant within the Zingiberaceae family, is native to the moist forests in southern regions and tropical zones of India [1,2,3]. Introduced to Guatemala from India between 1910 to 1914, it has flourished predominantly in the Alta Verapaz department [4,5]. By 1911, Coban municipality was recognized as an established local production and this was followed by expansion into the Southern Coast region of Guatemala [5]. However, a significant setback occurred in 1975 with the first report of Cardamom mosaic virus (CdMV) in the Quetzaltenango department [6]. The outbreak led to over a decade of crop devastation and compelled cultivators to shift their operations to Guatemala’s Northern Transversal Strip, an event that likely resulted in a constriction event affecting genetic diversity among Guatemalan cardamom populations.

Currently, the principal production of cardamom is geographically concentrated within the North Transversal Stripe, encompassing several departments, namely Alta Verapaz, Baja Verapaz, Izabal, Huehuetenango, and Quiché [7,8,9]. In 2011, a critical constriction occurred due to an infestation by the thrips species *Scriothrips cardamomi*, which at its apex, affected approximately 90% of plantations [10]. Despite these formidable challenges, Guatemala continues to be one of the foremost producers and exporters of cardamom. This is attributed largely to the spice’s scant inclusion in local market [11]. In 2022, it was reported that exports exceeded USD 450 million and up to 45 thousand tons per year [8]. The cardamom industry has become a cornerstone of economic stability and a source of employment for a multitude of smallholder farmers and their dependents [12,13].

Recent initiatives have focused on delineating the stakeholders engaged in the production of Guatemalan cardamom and elucidating the multifaceted processes within its value chain [13]. A conceptualized value chain model comprises four integral segments: procurement of raw materials, production, primary transformation, and commercialization. Accompanying this segmentation, a compendium of research prospects has been identified, targeting the augmentation of overall productivity. This includes investigative studies on the propagation of cardamom varieties among local farmers [13].

Studies conducted by Pöll between 1987 and 1990 in the northern and southern regions of Guatemala revealed 21 cardamom “types”, and the author highlights that they refrain from using the term “variety” as it has not been confirmed that these morphological variants are underpinned by a genetic basis. The primary differences among these varieties are related to the type of inflorescence, capsule shape, and leaf form [14]. This had been the sole study aimed at elucidating the diversity and establishing a classification criterion for the cardamom in Guatemala. On the other hand, other studies have been conducted in the Republic of India, such as the study conducted with varieties from the Indian Cardamom Research Institute (ICRI) carried out by Alam et al. [2] to evaluate three high-yielding varieties named ‘Valley Green’, ‘Palakuzhi’, and ‘ICRI’, which were derived from “natural cultivars” known as ‘Mysore’, ‘Malabar’, and ‘Vazhukka’. The principal morphological differences they found among these three varieties are related to the basis of panicles, capsule shape, plant height, tiller number, and seeds per capsule [2].

This study proposes the utilization of morphological traits published by the International Plant Genetic Resources Institute (IPGRI) to evaluate a sample of 288 cardamom accessions and select out of them outstanding plant material in the North Transversal Stripe of Guatemala. The method of comprehensive evaluation, previously employed by other authors for assessing and selecting germplasm of various crops, is proposed [15,16,17,18,19]. Comprehensive evaluation is a rigorous process that facilitates the examination and systematization of diverse morphological, phenotypic, physiological, and biochemical characteristics of germplasm through scoring [17,18,20,21]. The outcomes of this methodology enable the selection of excellent germplasm accessions, which may be utilized in the future for crop improvement and the conservation of genetic diversity.

## 2. Results

### 2.1. Diversity Analysis of Cardamom Morphological Traits

The 17 morphological traits used to characterize the diversity of cardamom germplasm, selected based on information published by the International Plant Genetic Resources Institute (IPGRI) (Table 1), showed a minimum value of the Shannon–Wiener genetic diversity index (H’) of 0.10 and a maximum of 2.02.

Among the 17 traits, two were quantitative variables: cluster internodal length and the number of capsules per cluster. These traits exhibited the highest diversity indices (1.68 and 2.02, respectively). Additionally, the coefficient of variation for these two traits was substantial (180.30% and 287.16%, respectively), indicating significant variation across observations. Conversely, the traits related to panicle type, stem base color, panicle branching, and panicle branching pattern had the lowest diversity indices (0.10, 0.14, 0.17, and 0.17, respectively). Regarding the remaining qualitative morphological traits, the highest diversity indices were observed for the number of stems per plant, panicle length, and the number of panicles per plant (1.33, 1.28, and 1.23, respectively), indicating moderate variation.

### 2.2. Principal Components of Cardamom Morphological Traits

To explain cardamom germplasm, we conducted a principal component analysis using the 17 morphological variables. This allowed us to select the most influential components through standardization and dimension reduction. Initially, we considered the first 14 principal components, which collectively explained 74.348% of the data (as observed in Table 2). However, the individual contribution of each component was relatively low, and the main traits in the first three principal components persisted in the other fourteen. Consequently, we proposed focusing on the first three principal components, which together covered 24.440% of the variance and had individual contributions exceeding 6%.

The first principal component exhibits an eigenvalue of 0.797, contributing 11.108% to the overall variance. The most influential traits in this component include plant height (when exceeding 3 m), plant height (between 2 and 3 m), the number of stems per plant (when exceeding 45), the number of panicles per plant (when exceeding 31), and the number of clusters per panicle (when fewer than 20 or between 21 and 30). Notably, this primary component is associated with numerical traits related to accession productivity.

The second principal component has an eigenvalue of 0.498, contributing 6.934%. This component is associated with traits related to glabra-type pubescence on the leaf, as well as capsule shape when it is globular and ovoid. The third principal component has an eigenvalue of 0.459, contributing 6.398%. In this component the glabra-type pubescence on the leaf persists and adds the rhizome color (light purple), and the capsule color (green and light green). This final component indicates that visual parameters, such as color, can be utilized to elucidate cardamom germplasm.

These same traits remain pertinent in the subsequent principal components up to the fourteenth, and some new ones are added, such as panicle length up to the ninth principal component, average internodal length in the eighth principal component, and the number of capsules per cluster in the fourteenth principal component.

### 2.3. Comprehensive Evaluation of Cardamom Morphological Traits and Identification of Excellent Accessions

A comprehensive evaluation analysis was conducted for (1) the first three principal components and (2) the initial fourteen principal components to corroborate the decision to select the first three as the most significant in explaining cardamom germplasm and to allow for the selection of outstanding accessions. For this analysis, the F-value was calculated following the method proposed by Hu et al. [16], and a ranking based on the highest F-value was determined allowing to select the top 14 selected accessions with F-values between 0.732 and 0.819 when the first three principal components were used (Table 3), which varied between 0.589 and 0.674 when the first 14 PCs were used. When comparing the top accessions generated using the first three principal components against the top accessions generated using the first 14 PCs, it can be observed that although they are not in the same position, the first seven accessions are present in both rankings. Subsequently, positions 9, 10, 17, 18, and 22 from the ranking created with the first 14 PCs (Appendix A) appear when considering the top 14 from the ranking created with the first three principal components.

Only two accessions selected from analyzing with the just three principal components were not chosen when considering all fourteen. Accession number GTM-14-20-066-001 differs from others by having fewer panicles per plant (<10) and shorter panicle length (<50 cm), whereas other remaining 13 accessions have between 11 to 20 panicles per plant and more than 50 cm panicle length (Appendix A). On the other hand, accession number GTM-14-20-077-001 distinguishes itself by having 20 capsules per cluster, surpassing the others with fewer than 20 capsules each (Appendix A). Furthermore, its ovoid-shaped capsule differs from the predominantly globular forms observed in the remaining selections. Given these few variations, it is reasonable to utilize only the three main components to explain germplasm variability and select superior cardamom accessions. Finally, the F-values derived from the first three principal component analyses ranged from 0.193 to 0.819, indicating the accessions with the most favorable and least favorable composite traits, respectively (Appendix A).

### 2.4. Regression Model for Identification of Key Cardamom Morphological Traits

Utilizing the F-values derived from the initial three principal components of each accession, alongside 17 variables, a stepwise linear regression was performed to identify a model that effectively characterizes key morphological characteristics in cardamom germplasm. The resulting equation, based on the selected model, is as follows:$$y=\;0.627\;-\;0.119x_{1}\;+\;0.044x_{2}\;-\;0.104x_{3}\;-\;0.070x_{4}\;+\;0.056x_{5}\;-\;0.036x_{6}\;+\;0.045x_{7}\;+\;0.034x_{8}\;\;-\;0.042x_{9}\;+\;0.036x_{10}\;-\;0.058x_{11}\;+\;0.040x_{12}$$

Here, the independent variables represent plant height (>3 m), capsule color (green), panicles per plant (>31), capsule shape (ovoid), panicle length (51–75 cm), rhizome color (light purple), clusters per panicle (<20), internodal length (1.6–2.5 cm), panicles per plant (21–30), capsules per cluster (≥13), capsule color (light green), and rhizome color (light green), respectively. This model exhibits a high correlation coefficient (R = 0.986) and a coefficient of determination (R^2^ = 0.972), indicating that these twelve independent variables can account for approximately 97.2% of the total variance observed in the F-value.

### 2.5. Hierarchical Cluster Analysis of Cardamom Morphological Traits

A hierarchical cluster analysis was conducted using the first three PCs to classify 288 cardamom accessions, which were categorized into six distinct groups (Figure 1). The diversity index assessed using the Shannon–Wiener genetic diversity index (H’) by considering the groups was relatively high, with a value of 1.28 (Table 4). The main properties of each group were as follows:Group 1 was characterized by accessions with a plant height greater than 3 m, with most accessions having 20 to 30 panicles and ovoid capsules. These accessions exhibited both light purple and light green rhizome colors.Group 2 stood out for its prevalence of ovoid and green capsules.Group 3 primarily clustered accessions with light purple rhizomes and light green capsules.Group 4 comprised accessions with a height predominantly between 2 and 3 m.Group 5 was characterized by accessions with glabrous leaves, panicle lengths ranging from 51 to 75 cm, ovoid capsules and of light green color capsules.Finally, Group 6 predominantly grouped plants with a height between 2 and 3 m and 20 to 30 clusters per panicle.

The majority of the 14 selected accessions, based on the F-value, were concentrated within groups IV and VI. Only one accession fell into group II (Figure 2). PC1 facilitated the segregation of cardamom germplasm groups. Specifically, Groups 1 and 2 predominantly occupied the left side of PC1, comprising accessions with heights exceeding 3 m. Meanwhile, Groups 4 and 6 were positioned on the right side of PC1, encompassing accessions with heights ranging from 2 to 3 m. Groups 3 and 5 exhibited uniform distribution along PC1.

### 2.6. Cardamom Varieties Based on Plant Height and Capsule Color

According to the plant descriptors provided by IPGRI, cardamom (*Elettaria cardamomum* (L.) Maton) can be divided into three putative foreign “natural cultivars”: (1) ‘Malabar’, (2) ‘Mysore’, and (3) ‘Vazhukka’. This third cultivar exhibits intermediate properties between the first two cultivars. Plant height plays a crucial role in determining the classification within the proposed varieties [22]. Accessions of cardamom were classified based on their height, with those below 3 m categorized as ‘Malabar’ and those exceeding 3 m as ‘Mysore’/‘Vazhukka’. Using the first three PCs, the graphical representation of this putative cultivar categorization (Figure 3a) differentiated the two groups on PC1, where plant height was a significant morphological trait (Table 2).

On the other hand, according to the literature and the results observed in this study, capsule color is a parameter that can also differentiate cardamom cultivars, specifically the germplasm evaluated in this study (Table 6) [22]. Cardamom accessions were classified as ‘Malabar’ for those with light green or yellow capsules, and as ‘Mysore’/‘Malabar’ for those with green or dark green capsules. The two groups based on capsule color were separated by PC3 (Figure 3b), which highlighted the significance of capsule color as a trait (Table 2).

### 2.7. Selection and Conservation of Excellent Cardamom Accessions

The comprehensive evaluation analysis allowed the selection of the best accessions that represent cardamom germplasm. However, when considering the putative foreign cardamom varieties, it became evident that the selection was biased towards ‘Malabar’ accessions based on plant height. This exclusion affected the accessions proposed as ‘Mysore’/’Vazhukka’ according to plant height criteria. Among the 14 previously selected accessions, 13 belong to the ‘Malabar’ variety (Table 5), resulting in an average diversity index H’ of 0.74. In a germplasm conservation strategy, a second comprehension analysis focused solely on ‘Mysore’/’Vazhukka’ accessions based on plant height criteria. This additional analysis led to the inclusion of 12 new excellent cardamom accessions in the list and the average diversity index H’ increased to 0.84. Furthermore, within this updated listing, there were at least two representative accessions from each dendrogram group.

Using the comprehensive evaluation analysis to select outstanding ‘Mysore’/’Vazhukka’ cardamom accessions, the following equation was obtained:y = 0.413 + 0.098*x*_1_ − 0.044*x*_2_ − 0.052*x*_3_ + 0.088*x*_4_ + 0.061*x*_5_ + 0.081*x*_6_ − 0.046*x*_7_ + 0.036*x*_8_ + 0.040*x*_9_ + 0.054*x*_10_ + 0.049*x*_11_ − 0.024*x*_12_,
which accounted the 98.4% of the observed variance with the F-value and considered the following independent variables: panicle length (51–75 cm), number of stems per plant (>45), number of clusters per panicle (21–30), capsule form (globose), rhizome color (light purple), internodal length (1.6–2.5 cm), number of panicles per plant (>31), number of capsules per cluster (≤8), capsule color (green), number of clusters per panicle (<20), internodal length (<1.5), and rhizome color (ligth green).

## 3. Discussion

### 3.1. Diversity in the Cardamom Germplasm by Morphological Traits

The deployment of morphological traits is essential for evaluating the genetic diversity within crop species. Such traits are instrumental in the identification and categorization of accessions, thereby laying the groundwork for the formulation of germplasm conservation strategies [23]. Additionally, analyses of genetic diversity are critical for evaluating the sustainability and diversification of agricultural systems [24,25,26]. In the specific context of cardamom cultivation in Guatemala, the use of these traits is vital for recognition existing diversity and elucidating the varieties under cultivation. This initial step is crucial for the subsequent selection of enhanced characteristics, which are targeted to improve crop productivity.

In this study, 17 morphological traits were assessed across 288 cardamom accessions from the Northern Transversal Strip of Guatemala, the region noted for its high productivity. The study yielded an average genetic diversity index of 0.86, with the minimum and maximum values being 0.10 and 2.02, respectively. The type of panicle was identified as the trait with the least genetic diversity, a result of the predominant occurrence of semi-erect panicles among the accessions. Hence, this particular trait may not be a significant differentiator among cardamom varieties. In contrast, the trait with the highest Shannon–Wiener index was the number of capsules per cluster, which was one of the two morphological variables gathered as quantitative data. This trait holds considerable significance in terms of productivity, especially since the capsule represents the marketable portion of the cardamom plant. Among the qualitative traits, number of stems per plant recorded the highest genetic diversity index (H’), with a value of 1.33. This finding suggests that it could serve as a metric for assessing plant productivity, indicating that certain accessions might be preferentially selected for traits that are indicative of higher productivity [27,28].

In the realm of cardamom research, there is a paucity of studies aimed at elucidating the genetic diversity of this spice. The investigation conducted by Anjali et al. in the Republic of India, which focused on the genetic diversity of cardamom, reported Shannon–Wiener’s Information Index values ranging from 0.23 to 0.54. [29]. However, it is noteworthy that this study was based on the analysis of ISSR molecular markers. In contrast, the research undertaken on Guatemalan cardamom demonstrated a higher level of diversity. Nonetheless, within the Zingiberaceae family, other genera such as *Amomum* have demonstrated diversity as measured by the Shannon–Wiener diversity index (H’), with values ranging from 1.71 to 2.22 [30]. Notably, the highest value reported is comparable to that observed in this study.

### 3.2. Comprehensive Evaluation of Cardamom Morphological Traits for the Identification of Excellent Germplasms

The comprehensive evaluation analysis has previously been employed by various authors to assess and select outstanding germplasm [15,16,17,18,19]. This method represents an integrated approach that combines principal component analysis (PCA) to identify the most influential variables on germplasm, standardizes data based on principal components, and utilizes a morphological composite value to score cardamom accessions and select the most promising ones [16,17]. In this study, the first three principal components accounted for 24.44% of the variation. It was observed that significant variables persisted up to the fourteenth principal component, with the next significant variable appearing in the twentieth PC. PC14 encompassed 74.35% of the germplasm explanation; however, some significant variables were repetitive from PC1, PC2, and PC3. The comprehensive evaluation analysis was conducted with both the first three PCs and the full set of fourteen PCs. The F-value derived from the morphological composite value analysis of the first three PCs enabled the selection of twelve accessions, which included the top seven positions from the analysis with fourteen PCs and other accessions up to rank number twenty-two. Consequently, the decision was made to utilize only the first three PC to elucidate the germplasm and select the superior accessions.

Various authors have reported the use of different numbers of principal components, achieving data explanation percentages ranging from 57.91% to 85.17%. Al-Naggar et al., utilizing morphological characteristics in maize, employed the first two principal components, which accounted for 57.91% of the variance [31]. Conversely, Devesh et al. used seven principal components, capturing approximately 66.22% of the variability, to evaluate yield, yield components, and quality traits of advanced wheat lines [32]. The study by Maji and Shaibu employed the first two principal components to describe the variation in rice germplasm, explaining 78% of the variance [33]. Similarly, Ramesh Kumar et al. selected the first six principal components, explaining 80.61% of the variation for the eggplant germplasm [34]. Li et al. reported the use of seven principal components to account for 85.17% of the total variance for a comprehensive evaluation analysis with persimmon [35]. In this study, it was possible to explain the data variation and select superior accessions using the first three principal components, with a maximum explained variance of 24.44%.

The first principal component analysis reveals that plant height is a significant trait to consider for elucidating the germplasm of cardamom. Other pertinent factors include the number of stems per plant (when exceeding 45), the number of panicles per plant (when exceeding 31), and the number of clusters per panicle (when fewer than 20 or up to 30). This primary component underscores the importance of variables linked to productivity within the germplasm, thus it is critically important to consider when selecting for enhanced productivity. The second principal component pertains to the phenology of leaves and capsules, specifically the glabrous-type pubescence of leaves and the shape of the capsules. The third principal component is also related to phenology, highlighting variables such as the rhizome color when it is light purple and the capsule color when it is green or light green. As visual parameters, the traits of the second and third principal components can be readily employed in the future to continue assessing the diversity of cardamom germplasm over the years.

Hu et al., utilizing a comprehensive evaluation analysis for the selection of maize accessions, assessed phenotypic traits and identified the traits most relevant for describing the variability of maize germplasm. Based on the selection of three principal components, these traits include ear height, ear height to plant height ratio, spike leaf width, leaf width of the upper ear, and effective spikes per plant [16]. The authors used these traits to establish key individual indicators for maize, and in this study we could establish two indicators related to productivity and leaf, capsules and rhizomes phenology. Maji and Shaibu employed morphological and agronomic traits to characterize and evaluate rice germplasm through principal component analysis; they identified plant height as one of the most significant variables, mirroring the findings of this study where plant height emerged as the most influential variable in the first principal component [33]. Additionally, they highlighted the relevance of variables such as leaf length, ligule length, days to 50% flowering, grain weight, number of grains per panicle, number of suckers at three weeks post-planting, leaf width, and number of unfilled grains for rice germplasm evaluation [33]. Pereira-Lorenzo et al., in their study on chestnut cultivars using morphological traits, identified nut size, shape, and sweetness as the most critical characteristics for this crop, as determined by principal component analysis. Each crop possesses unique traits that are essential for describing its germplasm [36]. Therefore, it is imperative to establish the most representative characteristics of Guatemalan cardamom germplasm.

It is crucial to emphasize that this study represents an initial effort to elucidate the key morphological characteristics that best describe cardamom germplasm. Although conducted with a single replicate, the study provides valuable preliminary insights. Future research should aim to evaluate the same 17 variables under different geo-environmental conditions to verify their stability over time and space. This is particularly important for identifying the optimal conditions for maximizing cardamom productivity. Additionally, future studies on morphological variables should be complemented by genetic evaluations using molecular markers, as demonstrated by Anjali et al. [29,37]. Furthermore, it has been proposed to confirm the same hierarchical groupings with microsatellite markers (Herrera et al., personal communication).

The F-value derived from the morphological composite value analysis using the first three principal components facilitated the selection of 14 accessions. In this study, the F-values obtained ranged from 0.193 to 0.819, signifying the accessions with the best and the worst composite traits, respectively (Appendix A). Comparatively, other studies have reported F-values ranging from a minimum of 0.089 to a maximum of 0.90 in maize, and a minimum of −1.50 and a maximum of 1.30 in apple trees [16,17]. The equation obtained from the linear regression analysis will enable the selection of these excellent accessions and relevant traits [16,17,38]. The linear equation favors the selection of 12 morphological traits classes including plants with less than 2 m in height, with a green rhizome, 20 or fewer panicles per plant, panicle lengths of 51 to 75 cm, fewer than 20 clusters per panicle, an internodal length between 1.6 to 2.5 cm, at least 13 capsules per cluster, and green capsules. In this equation, the trait of capsules per cluster will influence the selection of accessions with greater productivity, distinguishing those with more than 13 capsules per cluster. In studies that have conducted comprehensive evaluations, the linear equation comprises 10 variables with an R^2^ of 0.996 in the maize germplasm study, and 15 variables with an R^2^ of 0.978 in the apple germplasm study [16,17].

The 14 excellent accessions selected from the cardamom germplasm are primarily classified within groups IV and VI, with one accession belonging to group II, as determined by the hierarchical clustering analysis using the three main principal components. These groups are characterized by accessions ranging from 2 to 3 m in height, having 20 to 30 clusters per panicle, and featuring green, ovoid capsules. The number of selected accessions and the corresponding selection percentages vary across different crops and authors. Hu et al. selected 106 out of 192 maize accessions based on an F-value criterion greater than 0.5, resulting in a selection percentage of 55% [16]. Conversely, Tian et al. selected 10 out of 256 apple germplasm accessions, representing 4%, with these selections having an F-value greater than 0.79 [17]. Similarly, Pei-jun et al. selected 10 out of 166 tomato germplasm accessions, achieving a selection percentage of 6% [39]. Zheng et al. applied a selection criterion to 30% of the *Alnus cremastogyne* accessions, equating to 12 out of 40 studied accessions [40]. In this study, the 14 selected excellent accessions constitute approximately 5% of the cardamom germplasm.

### 3.3. Conservation of the Cardamom Germplasm

For Guatemala, an agricultural country distinguished by its high biodiversity, it is crucial to establish conservation strategies for plant materials [41]. It has been previously noted that there are 14 accessions deemed excellent for selection and use in genetic improvement and productivity enhancement strategies within the cardamom germplasm. This study also suggests an additional 12 accessions to preserve the crop’s diversity. Based on the literature review and given that Guatemalan cardamom originates from the Republic of India, it was proposed to include putative varieties known as ‘Malabar’, ‘Mysore’, and ‘Vazhukka’ from the Republic of India [2,22]. According to the semi-erect panicle type, most of the Guatemalan cardamom belongs to the ‘Vazhukka’ variety, which exhibits intermediate characteristics between ‘Malabar’ and ‘Mysore’ [2,3]. However, one of the main criteria for classifying this proposal of foreign varieties is the size of the plant. The ‘Malabar’ variety produce plants less than 3 m height, while the ‘Mysore’ and ‘Vazhukka’ varieties are more robust plants that reach 3 m in height [2,3].

Considering the plant height, 13 of the 14 excellent accessions of Guatemalan cardamom germplasm would be categorized as ‘Malabar’. Consequently, a second comprehensive evaluation analysis was conducted with accessions classified as ‘Mysore’/‘Vazhukka’. A second equation was derived from linear regression analysis, which will facilitate the selection of accessions from this proposed variety that have panicle lengths between 51 to 75 cm, more than 45 stems per plant, fewer than 30 clusters per panicle, globular capsules, rhizomes of light purple or light green color, an internodal length less than 2.5 cm, more than 31 panicles per plant, 8 or fewer capsules per cluster, and green capsules. Certain characteristics may be linked to the plant’s productive parameters, facilitating the selection of superior accessions of ‘Mysore’/‘Vazhukka’. This is underscored by Maji and Shaibu’s study, which also highlights the significance of analogous variables in maize in their principal component analysis [33]. Similarly, Tian et al. have demonstrated the critical role of parameters related to color phenology in the linear regression equation for identifying outstanding accessions [17].

The first accession selected through this second comprehensive evaluation analysis coincided with position number 14 from the initial selection. An additional 12 accessions were added to be considered in a list of cardamom accessions that allow for the conservation of germplasm. As evidence of this, the 26 accessions represent the six groups classified by the hierarchical clustering analysis. Moreover, the diversity index (H’) increased from 0.74 with the first 14 selected accessions to 0.84 with the 26 accessions selected for this conservation proposal.

Additionally, the color of the capsule is also a criterion for the classification of the proposed varieties of ‘Malabar’, ‘Mysore’, and ‘Vazhukka’ [22]. Therefore, plant height and capsule color serve as key indicators for putative foreign cultivars such as ‘Malabar’ and ‘Mysore’/‘Vazhukka,’ providing clear visual cues for classification. However, the criterion of height was considered more relevant as it is the variable with the most influence on the first principal component of this study. The capsule color variable was significant in the third principal component. It is crucial to emphasize that parameters related to the capsule, including shape, aroma, color, and physicochemical properties, have been identified as key determinants in the cardamom market [2,42]. Consequently, the capsule color highlighted in this study should be considered a significant variable in the criteria for selecting superior accessions and in conservation strategies, even though it is not the primary criterion for classifying varieties.

## 4. Materials and Methods

### 4.1. Plant Materials

This research was conducted across eleven productive regions within the Northern Transversal Strip of Guatemala, encompassing areas within Alta Verapaz, Quiché, and Izabal departments. These regions are recognized as the principal locales for cardamom cultivation in Guatemala [9]. The study utilized a sample of 288 cardamom accessions to gather morphological data. Specifically, samples included 29 accessions from Region I (Ixcán), 25 from Region II (Chajul and Nebaj), 60 from Region III (Cobán), 12 from Region IV (Chicamán and Uspantán), 18 from Region V (Chisec), 21 from Region VI (San Pedro Carchá), 27 from Region VII (Chajal and Fray Bartolomé de Las Casas), 48 from Region VIII (Cahabón and Lanquín), 21 from Region IX (Senahú, Tamahú, and Tucurú), 18 from Region X (Panzos, Santa Catarina La Tinta), and finally, 9 accessions were collected from Region XI (El Estor) (Figure 4).

### 4.2. Data Collection

Morphological parameters were selected from the information available at the International Plant Genetic Resources Institute (IPGRI) [22] (Table 6). This comprehensive dataset encompasses seventeen distinct characteristics, including stem base coloration, rhizome pigmentation, foliar morphology (shape and pubescence type), panicle characterization (number, length, panicle branching configuration, clusters density per panicle, and internodal distances), and capsules characterization (quantity, form, and color). The data were collected directly in the field at the accessions’ location as well as in the study conducted by Sunil et al. [43]. The decision to collect morphological data in situ was driven by the limited understanding of the optimal growth and development conditions for cardamom in Guatemala. The sampling sites were chosen to represent the primary cardamom-producing departments. In collaboration with local producers, the cardamom plants exhibiting the highest yield and production were selected, thereby ensuring the acquisition of superior accessions across different geographical regions. Appendix A provides an overview of the meteorological conditions, and the data collected by department within the study regions. It is noteworthy that, according to Cifuentes and Alonzo, the analysis of approximately 400 soil samples, including those from the collection sites for this study, indicates that soil fertility levels are quite similar across the study regions. Additionally, 88% of the soils in these areas are classified as clayey [44]. All morphological data collected for each accession were categorized into dichotomous outcomes of presence and absence for each observation [36].

### 4.3. Data Analysis

#### 4.3.1. Diversity Analysis

The data were sorted using Microsoft Excel 2024, version 16.86 (24060916). With the same software, genetic diversity index was assessed using the Shannon–Wiener genetic diversity index (H’). The following formula was employed: H’ = −ΣP_i_ × ln(P_i_) [16,17]. The two quantitative traits—average internodal length and number of capsules per cluster—were categorized into discrete levels based on their respective means (X) and standard deviations (σ). A stratification into ten levels was proposed, with an incremental difference of ±0.5σ between successive levels. This process began with level 1 [X_i_ < (X − 2σ)] and culminated at level 10 [X_i_ ≥ (X + 2σ)], as outlined in Hu et al. [16]. Herein, P_i_ denotes the frequency distribution of accessions within each defined level for numerical traits. Conversely, for qualitative traits, P_i_ signifies the observed frequency of each trait’s i-th accession manifestation. Lastly, ln represents the natural logarithm.

#### 4.3.2. Principal Component Analysis

Principal component analysis (PCA) was performed using IBM SPSS Statistics 29.0.02. employing a covariance matrix to ascertain the principal components [17,45].

#### 4.3.3. Hierarchical Cluster Analysis

Utilizing IBM SPSS Statistics 29.0.02., a hierarchical cluster analysis was conducted using the selected principal component analysis (PCA) as a basis. The linkage between groups was determined based on Euclidean distance, serving as the criterion for clustering [46].

#### 4.3.4. Comprehensive Evaluation of Morphological Traits

After conducting principal component analysis (PCA), a comprehensive evaluation method was developed to derive the morphological composite value (referred to as the “F-value”) using the formulas outlined by Hu et al. and Tian et al. [16,17]. This F-value was subsequently employed for identifying excellent accessions.

Next, a stepwise regression analysis was performed using IBM SPSS Statistics 29.0.02 software, incorporating both the F-value and the traits. From the various models, a stepwise regression equation was selected [16,17].

#### 4.3.5. Data Visualization

Data visualizations, including tables and figures, were generated using Microsoft Excel 2024, version 16.86 (24060916), and IBM SPSS Statistics 29.0.02. Additionally, maps representations detailing the geographic locations of the collected accessions were created with QGis, version 3.36.

## 5. Conclusions

The analyses of 17 morphological traits in 288 cardamom accessions showed a high value of diversity, with a maximum of H’ of 2.02, related to the germplasm of cardamom currently cultivated in Guatemala’s most productive region, the Northern Transversal Strip, and will serve as a starting point for further evaluating the diversity status of this crop. The comprehensive evaluation analysis allowed for the selection of 14 excellent cardamom germplasm accessions. Plant height, capsule color, number of panicles per plant, panicle length, rhizome color, number of clusters per panicle, cluster node length, and number of capsules per cluster were relevant in explaining the diversity of the germplasm. These 14 accessions were represented in three of the six groups formed from hierarchical clustering analysis. Considering this and the literature classification of cardamom into two proposed varieties named ‘Malabar’ and ‘Mysore’/‘Vazhukka’ based on plant height, another 12 accessions were selected from a second comprehensive evaluation analysis to represent the ‘Mysore’/‘Vazhukka’ variety and the other three groups obtained by the hierarchical clustering analysis. The selection of 26 accessions not only ensures the possession of excellent cardamom but also helps to conserve the diversity of the germplasm.

As one of the initial studies on cardamom germplasm from Guatemala’s most productive region, this research will facilitate more effective utilization of cardamom cultivation for future applications, particularly in enhancing productivity.

## Figures and Tables

**Figure 1 plants-13-02786-f001:**
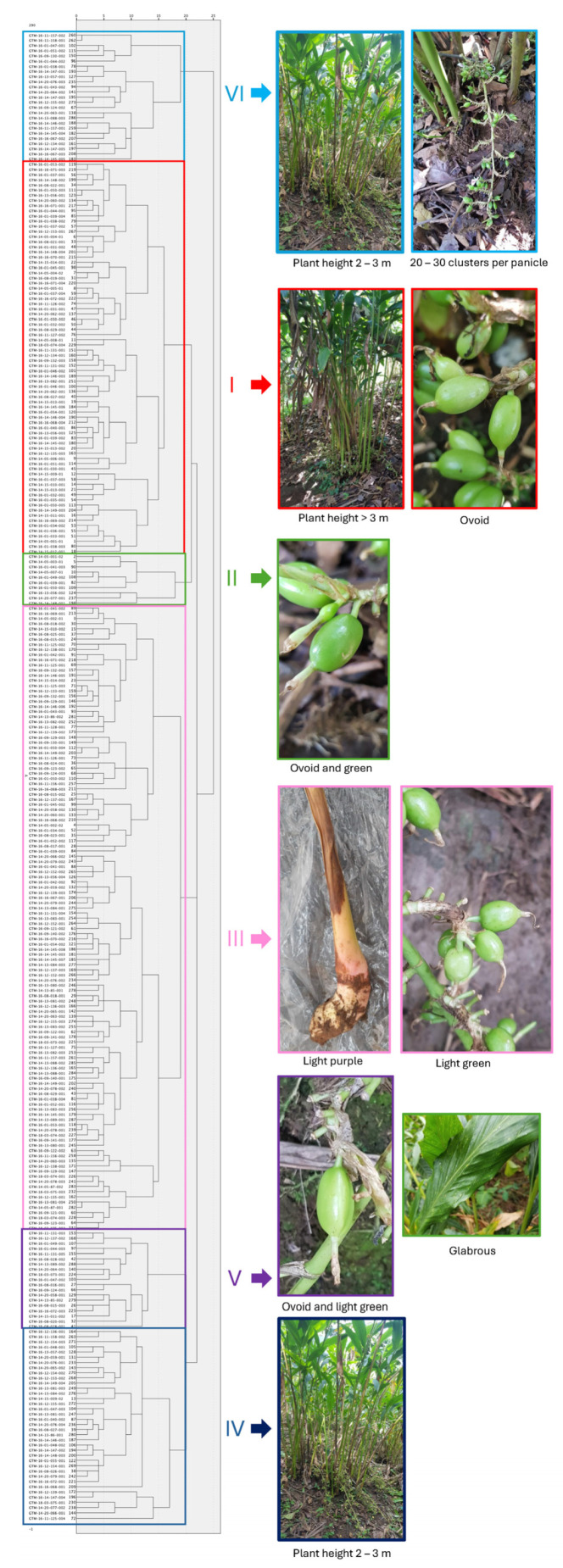
Hierarchical cluster analyses using the 3 PCs of the 17 morphological traits of 288 cardamom accessions from Guatemala.

**Figure 2 plants-13-02786-f002:**
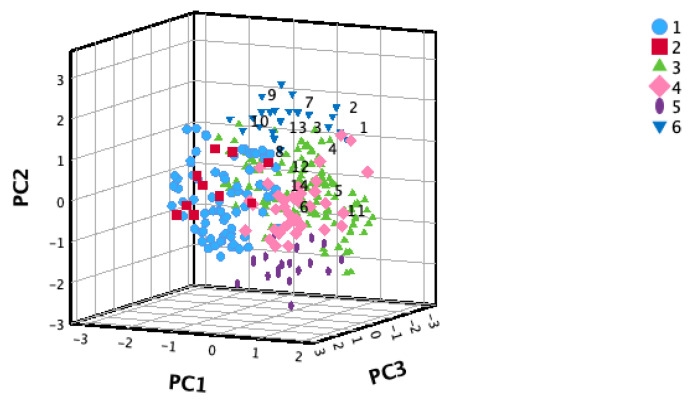
Selected cardamom accessions (1–14) classified into clusters (1–6) through hierarchical cluster analysis, utilizing the 3 principal components (PCs) derived from 17 morphological traits across 288 cardamom accessions.

**Figure 3 plants-13-02786-f003:**
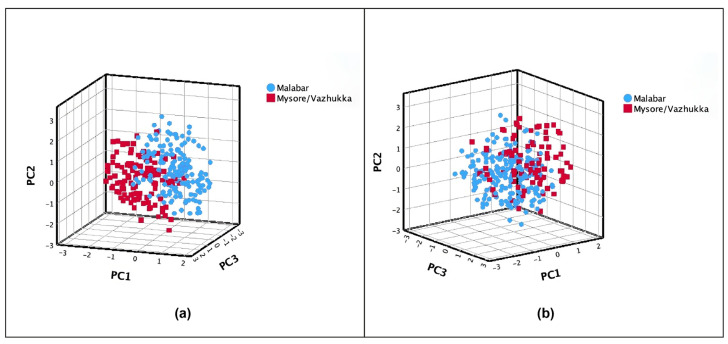
Principal component analysis (PC1–3) estimated with 17 morphological traits on 288 cardamom accessions classified in two putative foreign cultivars (‘Malabar’ and ‘Mysore’/’Vazhukka’) introduced in Guatemala according to (**a**) plant height and (**b**) color of the capsule.

**Figure 4 plants-13-02786-f004:**
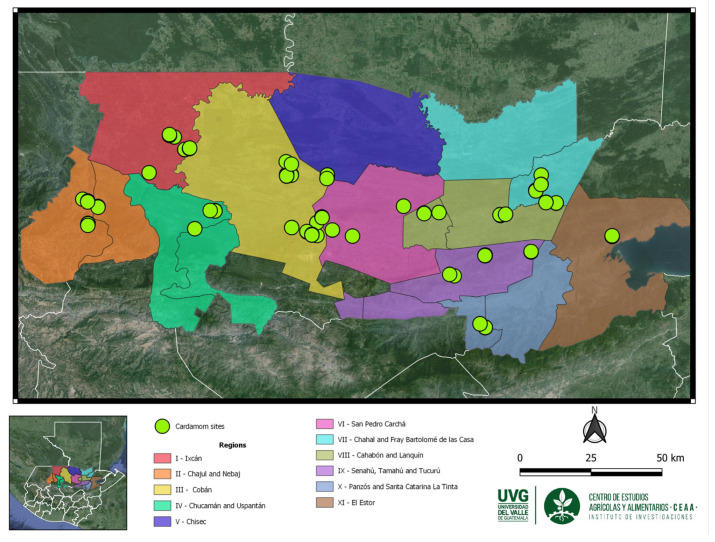
Collection sites of a sample of 288 accessions in the cardamom production regions at the Northern Transversal Strip of Guatemala.

**Table 1 plants-13-02786-t001:** Diversity analysis of cardamom with 17 morphological traits for 288 cardamom accessions from the North Transversal Stripe of Guatemala.

Trait	Morphological Classes (Observations)										Mean	Minimum	Maximum	SD	CV %	H’
	1	2	3	4	5	6	7	8	9	10						
Height of the Plant	2	167	119	-	-	-	-	-	-	-	-	-	-	-	-	0.72
Number of stems per plant	45	106	83	54	-	-	-	-	-	-	-	-	-	-	-	1.33
Stem color (base)	280	7	1	-	-	-	-	-	-	-	-	-	-	-	-	0.14
Rhizome color	32	168	86	1	1	-	-	-	-	-	-	-	-	-	-	0.96
Shape of the leaf	11	272	5	-	-	-	-	-	-	-	-	-	-	-	-	0.25
Pubescence of the leaf	141	81	66	-	-	-	-	-	-	-	-	-	-	-	-	1.04
Panicle type	2	283	3	-	-	-	-	-	-	-	-	-	-	-	-	0.10
Number of panicles per stem	17	257	14	-	-	-	-	-	-	-	-	-	-	-	-	0.42
Number of panicles per plant	20	56	86	126	-	-	-	-	-	-	-	-	-	-	-	1.23
Panicle length	34	112	95	47	-	-	-	-	-	-	-	-	-	-	-	1.28
Number of clusters per panicle	90	181	17	-	-	-	-	-	-	-	-	-	-	-	-	0.82
Cluster internodal length *	0	0	25	75	54	88	28	12	0	6	2.79	0.5	18	1.55	180.31	1.68
Panicle branching	12	276	-	-	-	-	-	-	-	-	-	-	-	-	-	0.17
Panicle branching pattern	12	276	-	-	-	-	-	-	-	-	-	-	-	-	-	0.17
Number of capsules per cluster *	3	12	42	32	66	62	20	33	10	8	10.75	3	25	3.74	287.16	2.02
Capsule form	13	73	202	-	-	-	-	-	-	-	-	-	-	-	-	0.74
Capsule color	11	76	13	188	-	-	-	-	-	-	-	-	-	-	-	0.89

* Quantitative variables. SD, standard deviation; CV, coefficient of variation; H’, Shannon–Wiener genetic diversity index.

**Table 2 plants-13-02786-t002:** Principal component analysis of the 17 morphological traits of 288 cardamom accessions.

Trait/Morphological Class		Principal Components (PC)													
		PC1	PC2	PC3	PC4	PC5	PC6	PC7	PC8	PC9	PC10	PC11	PC12	PC13	PC14
Height of the plant	<2 m	0.000	0.013	0.002	0.010	−0.006	0.005	0.002	−0.001	−0.004	0.001	0.004	−0.005	−0.006	0.002
	2–3 m	0.340	0.116	−0.055	−0.160	0.182	−0.044	−0.159	0.039	−0.096	0.054	0.049	0.005	−0.036	−0.043
	>3 m	−0.339	−0.129	0.053	0.151	−0.176	0.040	0.156	−0.038	0.100	−0.055	−0.052	−0.001	0.042	0.041
Number of stems per plant	<15	0.093	0.088	0.018	−0.011	−0.109	−0.021	−0.060	0.063	0.009	0.058	−0.137	−0.106	0.038	0.075
	16–30	0.176	0.000	−0.100	0.251	0.095	−0.049	0.198	0.051	−0.033	0.075	0.117	0.090	0.042	−0.092
	31–45	−0.053	−0.102	0.060	−0.147	0.016	−0.005	−0.170	−0.147	−0.011	−0.256	0.063	−0.024	−0.075	0.077
	>45	−0.216	0.014	0.023	−0.093	−0.001	0.075	0.032	0.034	0.035	0.124	−0.043	0.040	−0.004	−0.060
Stem color	Light green	0.013	−0.003	−0.023	0.014	0.000	−0.009	−0.012	−0.005	0.015	0.000	0.012	−0.007	−0.020	−0.013
	Green	−0.010	0.000	0.022	−0.014	0.009	0.008	0.015	0.011	−0.015	0.003	−0.015	0.009	0.023	0.014
	Yellow	−0.003	0.003	0.001	0.000	−0.009	0.001	−0.002	−0.006	0.000	−0.003	0.003	−0.002	−0.003	−0.002
Rhizome color	White	−0.048	0.000	0.064	0.046	0.039	−0.041	0.014	0.010	0.014	0.007	−0.035	−0.030	0.030	−0.019
	Light purple	0.082	−0.185	−0.270	−0.251	0.015	0.008	0.185	0.056	0.017	−0.043	0.030	−0.013	0.052	0.067
	Light green	−0.034	0.182	0.197	0.206	−0.048	0.032	−0.210	−0.063	−0.040	0.034	0.010	0.035	−0.087	−0.042
	Dark green	0.004	0.001	0.007	0.002	−0.005	0.001	0.007	0.002	0.001	0.000	−0.002	0.004	0.007	0.000
	Pale light purple	−0.004	0.002	0.002	−0.003	−0.001	−0.001	0.003	−0.005	0.008	0.003	−0.004	0.004	−0.002	−0.005
Shape of the leaf	Lanceolate	0.022	0.005	0.015	−0.017	0.016	−0.009	−0.004	0.016	0.017	−0.021	0.008	0.015	0.011	−0.003
	Oblong	−0.022	−0.004	−0.028	0.018	−0.014	−0.006	0.017	−0.024	−0.026	0.025	−0.018	−0.021	0.003	0.017
	Oval	0.000	−0.002	0.013	−0.001	−0.002	0.015	−0.014	0.008	0.009	−0.003	0.011	0.006	−0.013	−0.014
Pubescence of the leaf	Glabra	0.009	−0.240	0.220	−0.056	−0.080	−0.147	0.006	0.244	−0.200	−0.010	−0.068	0.000	−0.016	−0.073
	Dense	0.049	0.196	−0.151	−0.008	0.039	0.005	−0.042	−0.018	0.293	−0.059	0.002	0.074	0.031	−0.048
	Spaced	−0.058	0.044	−0.069	0.064	0.041	0.142	0.036	−0.226	−0.093	0.068	0.066	−0.074	−0.015	0.121
Panicle type	Creeping	−0.002	0.014	−0.003	0.001	−0.002	−0.003	−0.010	0.001	0.000	0.002	0.005	0.013	0.004	−0.003
	Semi-erect	0.008	−0.017	0.005	0.005	0.010	0.007	0.009	−0.007	−0.003	−0.005	0.001	−0.005	−0.013	0.013
	Erect	−0.006	0.002	−0.002	−0.006	−0.008	−0.004	0.001	0.005	0.003	0.004	−0.006	−0.007	0.009	−0.010
Number of panicles per stem	1	0.037	−0.008	−0.022	−0.006	−0.034	−0.019	0.023	0.037	0.000	0.059	0.015	0.010	−0.005	−0.021
	2	−0.017	−0.002	0.028	0.009	0.028	0.017	0.000	−0.037	0.003	−0.073	−0.021	−0.060	0.014	0.019
	>3	−0.020	0.010	−0.005	−0.003	0.006	0.003	−0.022	0.000	−0.003	0.015	0.006	0.050	−0.009	0.002
Number of panicles per plant	<10	0.076	0.017	0.014	−0.009	−0.076	0.028	−0.048	−0.001	−0.022	0.054	−0.052	−0.018	0.009	0.018
	11–20	0.128	0.100	0.003	0.093	0.011	−0.005	0.041	0.126	0.099	0.073	−0.081	−0.079	0.075	0.049
	21–30	0.125	−0.149	−0.030	0.165	0.056	−0.162	0.071	−0.150	−0.069	−0.178	0.066	−0.004	−0.076	0.008
	>31	−0.329	0.032	0.013	−0.249	0.009	0.139	−0.063	0.024	−0.008	0.051	0.067	0.101	−0.008	−0.075
Panicle length	<50 cm	0.116	−0.031	−0.037	0.055	−0.015	0.061	−0.090	−0.020	−0.065	0.049	−0.017	0.057	0.001	0.060
	51–75 cm	0.177	−0.083	0.188	−0.102	−0.108	−0.050	0.087	−0.076	0.245	0.025	0.085	−0.090	−0.096	−0.122
	76–100 cm	−0.164	0.144	−0.183	0.028	0.145	0.038	0.029	0.042	−0.132	−0.129	−0.212	−0.050	0.002	−0.009
	>100 cm	−0.129	−0.029	0.032	0.019	−0.022	−0.049	−0.026	0.054	−0.048	0.054	0.143	0.083	0.094	0.071
Number of clusters per panicle	<20	0.256	−0.182	0.065	0.047	−0.033	0.269	−0.061	−0.005	0.046	−0.012	−0.080	0.057	0.048	0.024
	20–30	−0.240	0.191	−0.045	−0.040	0.000	−0.300	0.036	0.000	−0.019	0.061	0.102	−0.111	−0.064	−0.006
	30–40	−0.017	−0.009	−0.021	−0.007	0.034	0.031	0.024	0.005	−0.027	−0.049	−0.023	0.054	0.016	−0.017
Average internodal length	≤1.5 cm	0.114	−0.084	−0.058	−0.010	0.016	0.045	−0.007	0.030	−0.008	0.105	0.091	0.022	−0.112	0.032
	1.6–2.5 cm	0.122	0.075	0.000	−0.110	−0.163	−0.195	0.024	−0.215	−0.026	−0.010	−0.217	0.074	0.096	−0.054
	2.6–3.5 cm	−0.099	0.021	0.064	0.091	0.187	0.107	−0.012	0.218	0.084	−0.190	0.014	−0.125	−0.042	−0.062
	≥3.5 cm	−0.138	−0.012	0.001	0.030	−0.040	0.046	−0.002	−0.033	−0.051	0.092	0.109	0.033	0.057	0.083
Panicle branching pattern	Proximal	0.011	−0.006	−0.001	0.018	0.048	−0.037	0.003	−0.010	0.003	−0.020	0.013	−0.016	0.049	0.001
	Absent	0.011	−0.006	−0.001	0.018	0.048	−0.037	0.003	−0.010	0.003	−0.020	0.013	−0.016	0.049	0.001
Number of capsules per cluster	≤8	0.027	−0.008	−0.152	0.014	−0.183	0.151	−0.014	−0.060	−0.118	0.022	0.058	−0.244	0.020	−0.191
	9–10	−0.074	−0.023	−0.024	0.029	0.086	−0.001	0.079	−0.042	−0.004	−0.005	−0.101	0.248	−0.157	−0.094
	11–12	0.027	−0.003	0.012	−0.013	−0.010	−0.076	0.000	0.108	0.090	0.069	−0.057	−0.038	−0.162	0.256
	≥13	0.020	0.034	0.164	−0.029	0.107	−0.075	−0.065	−0.006	0.032	−0.085	0.099	0.033	0.299	0.030
Capsule form	Ellipsoid	−0.008	0.018	0.013	−0.016	−0.011	0.020	0.004	0.033	−0.022	0.000	0.021	0.008	0.044	−0.009
	Globose	0.110	0.203	−0.027	−0.007	−0.244	0.043	0.069	0.094	−0.046	−0.154	0.075	0.081	−0.052	0.039
	Ovoid	−0.102	−0.221	0.014	0.023	0.256	−0.062	−0.073	−0.128	0.068	0.154	−0.096	−0.088	0.008	−0.030
Capsule color	Yellow	−0.025	−0.003	0.020	−0.030	0.004	0.010	0.015	−0.014	0.022	0.006	−0.009	−0.032	−0.016	−0.017
	Green	0.088	0.135	0.218	−0.072	0.103	0.090	0.222	−0.072	−0.087	0.016	−0.006	−0.010	0.011	0.042
	Dark green	0.005	0.029	0.003	−0.019	0.018	0.007	0.010	0.019	0.013	−0.020	0.001	−0.009	−0.033	−0.006
	Light green	−0.068	−0.161	−0.241	0.121	−0.125	−0.107	−0.248	0.068	0.051	−0.001	0.013	0.050	0.039	−0.019
Eigenvalue		0.797	0.498	0.459	0.420	0.419	0.394	0.374	0.354	0.313	0.306	0.275	0.263	0.243	0.222
Contribution rate (%)		11.108	6.934	6.398	5.847	5.835	5.493	5.217	4.928	4.360	4.261	3.825	3.660	3.383	3.098
Cumulative contribution rate (%)		11.108	18.042	24.440	30.287	36.122	41.614	46.831	51.759	56.120	60.381	64.206	67.866	71.250	74.348

**Table 3 plants-13-02786-t003:** Selected cardamom accessions based on the F-value among 288 accessions from Guatemala.

Accession	PC 1 to 3		PC 1 to 14	
	F-Value	Ranking	F-Value	Ranking
GTM-16-12-139-001	0.819	1	0.660	2
GTM-16-14-147-004	0.814	2	0.634	4
GTM-16-13-057-001	0.811	3	0.596	18
GTM-14-20-066-001	0.801	4	N/A	NS
GTM-14-20-065-002	0.784	5	0.589	22
GTM-16-12-154-002	0.778	6	0.627	7
GTM-14-20-076-003	0.776	7	0.656	3
GTM-14-20-077-001	0.774	8	N/A	NS
GTM-16-14-147-003	0.756	9	0.624	9
GTM-16-12-155-002	0.740	10	0.629	5
GTM-16-12-136-001	0.739	11	0.618	10
GTM-18-03-075-001	0.737	12	0.674	1
GTM-16-01-038-001	0.735	13	0.600	17
GTM-16-14-149-004	0.732	14	0.627	6

N/A, not applicable; NS, not selected.

**Table 4 plants-13-02786-t004:** Percentage (%) of accessions with a morphological class for the six groups of cardamom accessions obtained by a hierarchical cluster analysis.

Trait/Morphological Classes		Group by Hierarchical Cluster Analysis					
		1	2	3	4	5	6
n (number of accessions)		76	10	121	37	19	25
Global H’ for group classification		1.28					
Height of the plant	<2 m	0.00	0.00	0.00	0.00	0.00	8.00
	2–3 m	19.74	30.00	71.90	**86.49**	42.11	**88.00**
	>3 m	**80.26**	70.00	28.10	13.51	57.89	4.00
Number of stems per plant	<15	5.26	20.00	13.22	29.73	5.26	44.00
	16–30	15.79	10.00	50.41	40.54	42.11	36.00
	31–45	35.53	30.00	27.27	27.03	42.11	8.00
	>45	43.42	40.00	9.09	2.70	10.53	12.00
Stem color	Light green	93.42	90.00	99.17	97.30	100.00	100.00
	Green	5.26	10.00	0.83	2.70	0.00	0.00
	Yellow	1.32	0.00	0.00	0.00	0.00	0.00
Rhizome color	White	17.11	20.00	4.13	18.92	15.79	8.00
	Light purple	40.79	0.00	**83.47**	40.54	78.95	24.00
	Light green	40.79	**80.00**	12.40	37.84	5.26	68.00
	Dark green	0.00	0.00	0.00	2.70	0.00	0.00
	Pale light purple	1.32	0.00	0.00	0.00	0.00	0.00
Shape of the leaf	Lanceolate	1.32	10.00	3.31	8.11	5.26	4.00
	Oblong	97.37	80.00	95.04	91.89	94.74	92.00
	Oval	1.32	10.00	1.65	0.00	0.00	4.00
Pubescence of the leaf	Glabra	61.84	**80.00**	30.58	78.38	**94.74**	8.00
	Dense	11.84	10.00	41.32	10.81	0.00	68.00
	Spaced	26.32	10.00	28.10	10.81	5.26	24.00
Panicle type	Creeping	0.00	0.00	0.00	0.00	0.00	8.00
	Semi-erect	100.00	90.00	98.35	100.00	100.00	92.00
	Erect	0.00	10.00	1.65	0.00	0.00	0.00
Number of panicles per stem	1	3.95	0.00	7.44	8.11	10.53	0.00
	2	89.47	100.00	86.78	86.49	89.47	100.00
	>3	6.58	0.00	5.79	5.41	0.00	0.00
Number of panicles per plant	<10	0.00	10.00	7.44	18.92	0.00	12.00
	11–20	7.89	10.00	21.49	29.73	0.00	48.00
	21–30	17.11	10.00	35.54	29.73	78.95	12.00
	>31	75.00	70.00	35.54	21.62	21.05	28.00
Panicle length	<50 cm	1.32	0.00	20.66	13.51	10.53	4.00
	51–75 cm	26.32	60.00	26.45	78.38	**84.21**	36.00
	76–100 cm	34.21	10.00	42.98	5.41	5.26	52.00
	>100 cm	38.16	30.00	9.92	2.70	0.00	8.00
Number of clusters per panicle	<20	10.53	20.00	31.40	70.27	73.68	8.00
	20–30	82.89	70.00	60.33	27.03	26.32	**92.00**
	30–40	6.58	10.00	8.26	2.70	0.00	0.00
Average internodal length	≤1.5 cm	3.95	0.00	21.49	18.92	42.11	4.00
	1.6–2.5 cm	22.37	50.00	42.15	51.35	21.05	52.00
	2.6–3.5 cm	44.74	30.00	22.31	24.32	36.84	32.00
	≥3.5 cm	28.95	20.00	14.05	5.41	0.00	12.00
Panicle branching pattern	Proximal	0.00	10.00	7.44	2.70	5.26	0.00
	Absent	100.00	90.00	92.56	97.30	94.74	100.00
Number of capsules per cluster	≤8	18.42	10.00	46.28	18.92	36.84	16.00
	9–10	31.58	10.00	22.31	10.81	26.32	20.00
	11–12	26.32	0.00	13.22	29.73	26.32	40.00
	≥ 13	23.68	**80.00**	18.18	40.54	10.53	24.00
Capsule form	Ellipsoid	5.26	10.00	3.31	2.70	5.26	8.00
	Globose	11.84	10.00	26.45	35.14	0.00	72.00
	Ovoid	**82.89**	**80.00**	70.25	62.16	**94.74**	20.00
Capsule color	Yellow	6.58	10.00	1.65	8.11	0.00	0.00
	Green	21.05	**90.00**	8.26	62.16	15.79	60.00
	Dark green	5.26	0.00	4.96	2.70	0.00	8.00
	Light green	67.11	0.00	**85.12**	27.03	**84.21**	32.00

Bold font highlights the main properties of a group based on ≥80%. H’, Shannon–Wiener genetic diversity index.

**Table 5 plants-13-02786-t005:** Selected cardamom accessions based on the F-value for the two putative foreign cultivars of ‘Malabar’ and ‘Mysore’/’Vazhukka’ in two analyses, the first one including both cultivars and a second one with only accessions of ‘Mysore’/’Vazhukka’.

Cultivar by Plant Height	Accession	F-Value	Ranking for Each Cultivar	Group by Hierarchical Cluster Analysis
‘Malabar’	GTM-16-12-139-001	0.819	1	IV
‘Malabar’	GTM-16-14-147-004	0.814	2	IV
‘Malabar’	GTM-16-13-057-001	0.811	3	VI
‘Malabar’	GTM-14-20-066-001	0.801	4	IV
‘Malabar’	GTM-14-20-065-002	0.784	5	IV
‘Malabar’	GTM-16-12-154-002	0.778	6	IV
‘Malabar’	GTM-14-20-076-003	0.776	7	VI
‘Malabar’	GTM-14-20-077-001	0.774	8	II
‘Malabar’	GTM-16-14-147-003	0.756	9	VI
‘Malabar’	GTM-16-12-155-002	0.740	10	VI
‘Malabar’	GTM-16-12-136-001	0.739	11	IV
‘Malabar’	GTM-18-03-075-001	0.737	12	IV
‘Malabar’	GTM-16-01-038-001	0.735	13	VI
‘Mysore’/’Vazhukka‘	GTM-16-14-149-004 *	0.732/0.792	14/1	VI
‘Mysore’/’Vazhukka’	GTM-16-14-148-003	0.722	2	IV
‘Mysore’/’Vazhukka’	GTM-16-11-131-003	0.719	3	V
‘Mysore’/’Vazhukka’	GTM-14-15-013-001	0.718	4	I
‘Mysore’/’Vazhukka’	GTM-16-08-026-001	0.714	5	IV
‘Mysore’/’Vazhukka’	GTM-16-12-137-003	0.712	6	III
‘Mysore’/’Vazhukka’	GTM-16-01-053-001	0.708	7	III
‘Mysore’/’Vazhukka’	GTM-16-09-140-001	0.704	8	III
‘Mysore’/’Vazhukka’	GTM-16-08-022-001	0.686	9	I
‘Mysore’/’Vazhukka’	GTM-16-08-028-001	0.686	10	V
‘Mysore’/’Vazhukka’	GTM-16-12-138-002	0.685	11	III
‘Mysore’/’Vazhukka’	GTM-16-01-038-004	0.684	12	III
‘Mysore’/’Vazhukka’	GTM-16-14-148-001	0.675	13	II

* Accession selected with the two comprehensive evaluation analysis.

**Table 6 plants-13-02786-t006:** Morphological traits (17) classes used for the study selected from the International Plant Genetic Resources Institute (IPGRI) [22].

Trait and IPGRI Code	Morphological Classes				
	1	2	3	4	5
7.1.4 Height of the plant	<2 m	2–3 m	>3 m	-	-
7.1.5 Number of stems per plant	<15	16–30	31–45	>45	-
7.1.6 Stem color (base)	Light green	Green	Yellow	-	-
7.1.8 Rhizome color	White	Light purple	Light green	Dark green	Pale light purple
7.1.10 Shape of the leaf	Lanceolate	Oblong	Oval	-	-
7.1.12 Pubescence of the leaf	Glabra	Dense	Spaced	-	-
7.2.4 Panicle type	Creeping	Semi-erect	Erect	-	-
7.2.6 Number of panicles per stem	1	2	>3	-	-
7.2.5 Number of panicles per plant	<10	11–20	21–30	>31	-
7.2.7 Panicle length (average of 5)	<50 cm	51–75 cm	76–100 cm	>100 cm	-
7.2.8 Number of clusters per panicle (average 5)	<20	20–30	30–40	-	-
7.2.10 Cluster internodal length	≤1.5 cm	1.6–2.5 cm	2.6–3.5 cm	≥3.5 cm	-
7.2.11 Panicle branching	Presence	Absence	-	-	-
7.2.11.1 Panicle branching pattern	Proximal	Not branching pattern	-	-	-
7.2.20 Number of capsules per cluster	≤8	9–10	11–12	≥13	-
7.2.21 Capsule form	Ellipsoid	Globose	Ovoid	-	-
7.2.23 Capsule color	Yellow	Green	Dark green	Light green	-

## Data Availability

The data are contained within the article or Appendix A.

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
