# Peer review of "Comprehensive Evaluation and Selection of Cardamom (Elettaria cardamomum (L.) Maton) Germplasm Using Morphological Traits"

_plants, 2024, doi:10.3390/plants13192786_

Round 1

Reviewer 1 Report

Comments and Suggestions for Authors

The paper investigates the morphological diversity of 288 cardamom accessions in Guatemala to identify key traits for selecting high-quality germplasm. Using principal component analysis (PCA), hierarchical clustering, and regression analysis, it reveals distinct germplasm groups and highlights traits such as plant height, capsule color, and panicle characteristics. It proposes a conservation strategy for foreign varieties and emphasizes the importance of these traits in enhancing productivity and conservation. Overall, the text of the manuscript is fairly well-organized but there are some areas that require revision for clarity. I hope my comments are helpful to the authors in revising their manuscript.

1. Environmental Variability: The influence of environmental conditions on the morphological traits is not addressed, potentially affecting the generalizability of the findings.

2. Field Validation: The lack of field validation means the practical applicability of the selected traits for improving productivity and conservation remains uncertain.

3. Statistical Methods: The choice of PCA and the specific number of principal components used may not fully encompass the variability within the germplasm.

4. Figure 3 is proposed to provide a clearer resolution.

5. Some sentences lack appropriate articles or prepositions, such as in “the importance of these traits in enhancing productivity and conservation,” which can be slightly awkward.

6. Repetitive Language: Words like "morphological traits" are repeated often without much variation, which could be improved for readability.

Moreover, it would be nice if you could find a way to improve readability of these small and crowded sub-figures,Or just choose some important relationships to present in the main text.

Comments on the Quality of English Language

Minor editing of English language required.

Reviewer 2 Report

Comments and Suggestions for Authors

Investigating and identifying species' morphological traits are crucial for the breeding of superior germplasm. This manuscript conducted a comprehensive evaluation of 17 morphological traits across 288 Cardamom germplasm. This research is highly significant for the breeding of Cardamom species. The manuscript is well-organized and makes an important contribution to the field of crop genetic breeding research.

However, there is a relevant issue that the authors need to address or elaborate on in the manuscript: Crop phenotypic traits are easily influenced by environmental conditions such as soil nutrient status and climatic factors, leading to variations in the morphological traits of the same germplasm under different ecological conditions. The authors are requested to provide a detailed explanation in the Materials and Methods section regarding the layout, experimental repetitions, and environmental conditions for the 288 Cardamom germplasm.

How was the stability of the 17 phenotypic traits assessed in this manuscript? It is also recommended that the manuscript should discuss the stability of these morphological traits in the Discussion section.

Round 2

Reviewer 1 Report

Comments and Suggestions for Authors

The authors have answered all the questions I rasied.

Reviewer 2 Report

Comments and Suggestions for Authors

I am satisfied with the authors' responses and the revisions made to the manuscript. I suggest accepting the potential publication of the manuscript.